Cryptocentrus steinhardti (Actinopterygii; Gobiidae): a new species of shrimp-goby, and a new invasive to the Mediterranean Sea

http://orcid.org/0000-0003-1597-269X Goren Menachem 1
http://orcid.org/0000-0002-4834-3091 Stern Nir 2 nirstern@ocean.org.il
1 School of Zoology, The George S. Wise Faculty of Life and Sciences and the Steinhardt Museum of Natural History, Tel Aviv University , Tel Aviv , Israel
2 Marine Biology, Israeli Oceanographic and Limnological Research Institute , Haifa , Israel
Justine Jean-Lou
Electronic publication date: 2021 Sep 28
Publication date: 2021
Volume: 9
Electronic Location ID: e12136
Received 2021 Jan 6; Accepted 2021 Aug 18
Copyright: © 2021 Goren and Stern
Copyright year: 2021
Copyright holder: Goren and Stern
License: This is an open access article distributed under the terms of the Creative Commons Attribution License, which permits unrestricted use, distribution, reproduction and adaptation in any medium and for any purpose provided that it is properly attributed. For attribution, the original author(s), title, publication source (PeerJ) and either DOI or URL of the article must be cited.
License URL: https://creativecommons.org/licenses/by/4.0/

Keywords: Integrative taxonomy, Lessepsian migration, Shrimp-goby, New species, Mediterranean Sea

Funding: Israeli Ministry of Environmental Protection This work was supported by the Israeli Ministry of Environmental Protection as part of the national monitoring project of the Israeli Mediterranean Sea. The funders had no role in study design, data collection and analysis, decision to publish, or preparation of the manuscript.

==============================
A new species of shrimp-goby was collected at depths of 60–80 m off the southern Israeli Mediterranean coast. A unique ‘DNA barcoding’ signature (mtDNA COI and Cytb) revealed that it differs from any other previously bar-coded goby species clustered phylogenetically with the shrimp-gobies group, in which Cryptocentrus is the most speciose genus. A morphological study supported the assignment of the fish to Cryptocentrus and differentiated the new species from its congeners. The species is described here as Cryptocentrus steinhardti n. sp. However, the present phylogenetic analysis demonstrates a paraphyly of Cryptocentrus and emphasizes the need for revision of the genus based on integrating morphological and genetic characteristics. This finding constitutes the third record of an invasive shrimp goby in the Mediterranean Sea. An intriguing ecological issue arises regarding the possible formation of a fish-shrimp symbiosis in a newly invaded territory. Describing an alien tropical species in the Mediterranean prior to its discovery in native distribution is an unusual event, although not the first such case. Several similar examples are provided in the present article.

Introduction

Since the opening of the Suez Canal in 1869 more than 400 multicellular non-native species of Red Sea origin, including ca. A total of 100 fish species, have been found along the Israeli Mediterranean coast (Galil et al., 2020). Among this diverse invasive fauna there are two species of shrimp-gobies: Vanderhorstia mertensi Klausewitz, 1974 (Goren, Stern & Galil, 2013) and Cryptocentrus caeruleopunctatus (Rüppell, 1830) (Rothman & Goren, 2015). These species are part of a group of near-reef fishes that inhabit sandy and silty habitats and display a remarkable mutualism with burrowing alpheid shrimp, exchanging burrow construction capabilities and sentinel services (Karplus & Thompson, 2011). Common throughout the tropics, this unique fish-shrimp association is documented from over 100 fish species, that belong to 11 valid genera of gobies: Amblyeleotris Bleeker, 1874; Cryptocentrus Valenciennes (ex Ehrenberg) in Cuvier & Valenciennes, 1837; Cryptocentroides Popta, 1922; Ctenogobiops Smith, 1959; Lotilia Klausewitz, 1960; Mahidolia Smith, 1932; Myersina Herre, 1934; Psilogobius Baldwin, 1972; Stonogobiops Polunin & Lubbock, 1977; Tomiyamichthys Smith, 1956 and Vanderhorstia Smith, 1949 (Karplus, 2014; Ray, Mohapatra & Larson, 2018). An additional genus, Flabelligobius Smith, 1956, is considered a synonym of Tomiyamichthys (Hoese, Shibukawa & Johnson, 2016; Fricke, Eschmeyer & van der Laan, 2021).

During cruises to sample the benthic biota off Ashdod (southern Israel, Mediterranean coast), three specimens of an unknown shrimp-goby were collected at depths of 60 and 80 m by a bottom trawl net. An integrated study using both traditional practices and molecular taxonomy indicated that these fish belong to an undescribed species of Cryptocentrus genus.

Materials & methods

Fish specimens were collected from the southern sandy coast of the Israeli Mediterranean by a commercial 240 hp F/V bottom trawler. The fish were preserved in 70% alcohol and stored at the fish collection of The Steinhardt Museum of Natural History, Tel-Aviv University (SMNHTAU). Muscle tissue samples were taken from fresh specimens for genetic analyses and preserved in 96% alcohol.

For counts and measurements of meristic characteristics we followed Allen, Erdmann & Brooks (2018).

Genetic analysis

Total genomic DNA was extracted from the three specimens using a micro tissue genomic DNA isolation kit following the manufacturer’s protocol (AMBRD Laboratories, Turkey). Next, ca. 50 ng of template DNA were used to amplify a 651 bp fragment of the mitochondrial cytochrome c oxidase subunit I gene (COI) and 467 bp of the mitochondrial Cytochrome b (Cytb). Primers and PCR reactions are detailed in Table S1. The contiguous sequences of both genes, including measurements, a photo and trace files, were uploaded to the BOLD system at www.v4.boldsystems.org under the BIM project (Biota of the Israeli Mediterranean) with BOLD Sample IDs: BIM769-20 for the holotype and BIM534-17 and BIM770-20 for the two paratypes. Due to the absence of Cytb sequences for other shrimp-associated gobies, only the COI gene was used to explore the phylogeny of this group. For this purpose, 107 previously published sequences belonging to ten putative genera were mined from BOLD and NCBI and aligned using ClustalW, with a single sequence of Gobius niger as an outgroup (Table S2). The genetic vouchers were included in the dataset only if they indicated a precise sampling locality and an unambiguous association with a Barcode Index Number (BIN) of their corresponding taxonomic identifications. Sequences of Cryptocentrus yatsui, for example, were excluded from the analyses since they shared a BIN with the gobies Oligolepis formosanus and Redigobius bikolanus (BIN:BOLD:ADB4723). The best model test for nucleotide substitution was verified for the aligned dataset using jModelTest ver. 2.1.10 (Darriba et al., 2012) under the Akaike Information Criterion (AIC). Finally, Maximum Likelihood phylogenetic reconstruction was computed using the online program NGPhylogeny.fr (Lemoine et al., 2019) and the model HKY85+G+I with 500 replicates.

Nomenclatural acts

The electronic edition of this article in Portable Document Format (PDF), will represent a published work according to the International Commission on Zoological Nomenclature (ICZN), and hence the new name contained in the electronic edition is effectively published under this Code. This published work and the nomenclatural acts it contains have been registered in ZooBank, the online registration system for the ICZN. The ZooBank LSIDs (Life Science Identifiers) can be resolved and the associated information viewed through any standard web browser by appending the LSID to the prefix http://zoobank.org/. The LSID for this publication is: urn:lsid:zoobank.org:pub:B5279F4D-F5BC-454D-9ED8-3E2A13C69EAE. The online edition of this work is archived and available from the following digital repositories: PeerJ, PubMed Central, and CLOCKSS.

Results

Cryptocentrussteinhardti n. sp.

Steinhardt’s shrimp-goby

Figures 1, 210.7717/peerj.12136/fig-1 Figure 1 Holotype of Cryptocentrus steinhardti n.sp.

(SMNH P-16037).

Figure 2 Cephalic sensory system Cryptocentrus steinhardti.

NP, nasal pore; AIO, anterior interorbital pore; AO, anterior oculoscapular canal; PIO, posterior interorbital pore; PO, post orbital pore; POP, preopercolar pores; GO, Lower margin of gill opening.

Holotype: SMNH P-16037 [BOLD voucher BIM769-20], 81.9 mm total length (TL), Ashdod, Israel (31° 44.835 N, 34° 24.787 E), depth 80 m, 8 January, 2018, 19:45, coll. N. Stern.

Paratypes: SMNH P-14556 [BOLD voucher BIM534-17], 71.5 mm TL, Ashdod, Israel (31° 45.202 N, 34° 27.036 E), depth 60 m, 12 February, 2012, night, coll. N. Stern; SMNH P-16038 [BOLD voucher BIM770-20], 72.8 mm TL, Ashdod, Israel (31° 45.589 N, 34° 27.282 E), depth 60 m, 11 December, 2016, 19:45, coll. N. Stern.

Diagnosis

A Cryptocentrus species with 59–61 rows of cycloid scales along the body, 20–21 pre-dorsal scales, reaching ca. A total of 3/4 of the distance to eye (Fig. 2) and 19–21 transverse rows. Scales cover abdomen and prepelvic region. No scales on pectoral-fin base. First dorsal fin with six spines; second dorsal fin with a single spine and 10–11 segmented rays (last one branched). Anal fin with one spine and 9–10 segmented rays (last one branched). Pectoral fins with 15 rays. Pelvic fins completely united, with a well-developed fraenum. Caudal fin with 17 segmented rays, 13 of them branched.

Gill rakers: 10–11 on first gill arch, two on upper limb, one at the angle, and 7–8 on lower limb, the posterior three rakers are very short. Head sensory papillae in transverse pattern (sensu Miller, 1986).

Description

Body elongate and compressed. Upper profile of head convex. Mouth oblique. Maxilla extending to below the posterior margin of eye. Upper jaw with outer row of 16 caniniform teeth (eight on each side of the jaw) curved backward. Teeth in inner 1–2 rows small, pointed, curved backward. Lower jaw with outer 2–3 rows of small caniniform teeth, curved backward. Internal teeth in a single row of six large canines (three on either side of the jaw). No teeth on vomer. Tongue rounded.

Gill opening moderate, extending forward to below posterior margin of preopercle, restricted by a membrane on lower part (Fig. 2). The membranes of left and right sides are completely separate. Lower margin of opercle intersect at isthmus. Gill membrane connected to side of isthmus. Gill rakers short, 10–11 rakers on outer arch, two of them on upper limb, one at the angle, and 7–8 on lower limb, the posterior three rakers are very short. Anterior nostril, a tube, close above upper lip. Posterior nostril, a pore, in front of eye.

Scales: Body covered with cycloid scales, including abdomen and prepelvic region; pectoral fin base naked; 59–61 scales in longitudinal series; 20–21 mid-predorsal scales reaching ca. A total of 3/4 of the distance between dorsal fin and interorbital; 19–21 series of scales from origin of first dorsal fin to mid-abdomen.

Fins: First dorsal fin with six spines, third and fourth spines elongate, reaching the third ray of second dorsal fin. Second dorsal fin with a single spine and 10–11 segmented rays (last one branched). Rays long, the last three reach the caudal fin. Anal fin with one spine and 9–10 segmented rays (last one branched). Pectoral fins with 14–15 rays. Pelvic fins completely united to form a disc, with a well-developed fraenum. Caudal fin with 17 segmented rays, 13 of them branched.

Selected meristic characteristics and proportions are given in Table 1.

Table 1 Selected meristic characteristics and proportions (measurements in mm; proportion in %).

Measurements and counts	Fish catalogue number	
SMNHTUA 16037 (Holotype)	SMNHTAU 16038
(Paratype)	SMNHTAU 14556
(Paratype)	
Total length	81.9	72.8	71.46	
Standard length	58.5	51.1	49.95	
Head length	14.8	14.2	12.9	
Body depth	8.9	7.8	7.6	
Head width	5.8	5.4	5.1	
Eye diameter	3.44	4	3.6	
Interorbital	1.1	1.1	1	
Distance snout to origin of first dorsal fin	18	16.9	14.8	
Distance snout to origin of second dorsal fin	30.2	29.2	25.8	
Distance snout to origin of anal fin	33.4	30.5	28	
No. of scale series along the body	61	59	60	
No. of scale in transversal series	20	21	19	
No. pre-dorsal scales	21	20	20	
No. of spines in first dorsal fin	6	6	6	
No. of spines/segmented rays in second dorsal fin	I + 11	I + 10	I +1 0	
No. of spines/segmented rays in anal fin	I + 10	I + 9	I + 9	
No. of rays in pectoral fin (left side)	15	15	15	
No. of caudal rays	17	17	17	
Count of gill-rakers on upper arch	2	2	2	
Count of gill-rakers on upper arch	8	7	7	
Count of gill-rakers at arch angle	1	1	1	
Total count of gill-rakers	11	10	10	
Proportions (in %)				
Standard length of total length	71.4	70.2	69.9	
Head length of standard length	25.3	27.8	25.8	
Body depth of standard length	15.2	15.3	15.2	
Eye diameter of head length	23.2	28.2	27.9	
Interorbital space of head length	7.4	7.7	7.8	
Distance snout to origin of first dorsal fin	30.8	33.1	29.6	
Distance snout to origin of second dorsal fin	51.6	57.1	51.7	
Distance snout to origin of anal fin	57.1	59.7	56.1	

Cephalic sensory system: The skin of the head of all three type specimens was damaged in the commercial trawl net, hindering detection of the cephalic canal and papillae. Figure 2 presents the cephalic system of the specimen in the best condition (holotype).

Nasal pores (pair) in front of eye, close to posterior nostril opening. Anterior interorbital pore (single) above anterior margin of eye. Posterior interorbital pore above 1/6 posterior of eye. Postorbital pores (pair) above posterior margin of eye. A total of three pores in anterior oculoscapular canal. Posterior canal could not be detected (or does not exist). A total of two preopercular pores. Papillae on head arranged in a transverse pattern (Fig. 2).

Color (preserved): Body yellow with dark brown pigmentation that becomes denser on back and head. A total of three irregular wide darker bars on each side of body: the first bar under 1st dorsal fin and second and third bars under anterior and posterior parts of 2nd dorsal fin. Brown scattered spots on side of body in between broad bars. Chin with dark dense pigmentation. Distal half of first dorsal and anal fins’ membranes are black.

Genetic analysis

Comparing the genetic sequences of both COI and Cytb with previously published data revealed major differences to any other known gobies, with minimum distances in COI of 18.77% and 18.54% of nucleotide diversity between the new species and Cryptocentrus albidorsus and Stonogobiops xanthorhinica (BOLD vouchers GBGCA2109-13 and GBGCA2095-13, respectively) (Table 2), and 12.85% in Cytb differences between C. cinctus (NCBI voucher MT199211). Phylogenetic reconstruction of all available shrimp-associated gobies, incorporating for the first time representatives from the genera Lotilia, Myersina and Psilogobius, has revealed a basal separation between two groups of shrimp-gobies, in accordance with the suggestion by Thacker & Roje (2011): silt shrimp-gobies, which include our newly described species, and reef shrimp-gobies. Nevertheless, the poorly supported internal nodes within the tree emphasizes a systematic conundrum within this group (Fig. 3). C. steinhardti shares a branch with the genus Lotilia and other Cryptocentrus spp., though with low support for its generic assignment in terms of mtDNA phylogeny (Fig. 3).

Figure 3 ML phylogenetic analysis of all available COI sequences of shrimp-gobies.

Numbers above nodes are >50 bootstrap values; in red–the new species described in this study; in parentheses–number of sequences for each species. Further information for this dataset is provided in Table S2.

Table 2 Genetic relationships, in %, across all available COI sequences of shrimp-associated gobies.

In parentheses, no. of sequences for each species; below diagonal, pairwise genetic distances; above diagonal its standard errors. In red, values for Cryptocentrus steinhardti.

	1	2	3	4	5	6	7	8	9	10	11	12	13	14	15	16	17	18	19	20	21	22	23	24	25	26	27	28	29	30	31	
1. Cryptocentrus steinhardti (3)		2.09	2.50	2.70	2.49	2.97	2.63	2.80	2.19	2.31	2.88	2.77	2.29	2.50	2.85	2.72	2.98	2.97	3.38	2.79	2.99	3.04	2.65	2.76	2.90	2.83	2.66	2.28	2.93	2.90	3.47	
2. Cryptocentrus albidorsus (1)	18.77		2.56	3.04	3.12	3.13	2.64	2.39	2.13	1.97	1.57	1.49	1.75	3.04	2.49	2.67	3.44	2.73	3.28	3.20	3.46	2.85	2.80	3.20	3.07	3.59	3.29	3.01	2.75	3.12	3.12	
3. Cryptocentrus caeruleomaculatus (2)	22.37	17.71		3.11	3.06	2.75	2.66	2.28	2.08	2.04	1.98	2.69	2.18	3.01	2.48	3.12	3.26	2.89	3.65	3.56	2.91	2.92	2.57	3.22	3.00	3.07	3.18	3.13	3.20	2.72	2.59	
4. Cryptocentrus cebuanus (3)	22.83	23.78	23.70		2.57	1.83	1.89	3.22	2.83	3.29	3.02	3.50	3.00	0.92	3.12	2.96	3.85	3.03	2.99	3.44	3.45	3.39	3.12	2.54	3.87	3.37	3.43	2.90	3.61	3.48	3.18	
5. Cryptocentrus cinctus (12)	21.03	25.33	27.14	19.73		2.37	2.48	2.80	3.08	3.32	2.90	3.38	3.15	2.33	3.14	2.65	3.39	3.10	2.84	3.35	3.52	3.26	2.97	2.66	3.14	2.86	2.53	3.00	3.02	3.11	3.53	
6. Cryptocentrus cryptocentrus (1)	25.33	23.83	21.93	13.20	17.18		2.41	3.28	3.15	3.24	2.62	3.77	3.33	2.16	3.48	3.60	3.55	3.61	3.18	3.59	3.79	3.37	3.14	2.83	3.13	2.79	2.99	2.96	3.74	3.70	2.98	
7. Cryptocentrus cyanotaenia (3)	23.15	22.94	22.04	15.04	19.77	16.39		2.85	2.60	3.03	2.48	3.07	2.71	1.92	2.91	2.84	4.21	2.67	3.35	3.48	3.45	3.06	2.45	2.72	3.03	3.73	3.59	3.23	3.39	3.51	3.09	
8. Cryptocentrus inexplicatus (3)	23.96	16.53	15.82	25.27	24.66	25.73	22.65		2.16	2.05	2.54	2.28	2.56	3.13	1.95	2.86	3.34	3.13	3.20	3.05	2.94	2.80	2.80	3.22	3.18	3.29	3.37	3.04	2.56	2.80	2.89	
9. Cryptocentrus leptocephalus (19)	18.46	16.44	15.35	20.92	23.84	22.39	20.62	16.99		2.05	2.45	2.52	2.04	2.82	2.42	2.75	3.33	3.22	3.13	3.53	2.90	2.86	2.81	2.98	3.13	2.85	2.72	2.51	3.12	2.74	2.88	
10. Cryptocentrus lutheri (4)	20.09	14.23	15.10	24.79	26.69	25.24	24.84	15.67	13.90		2.16	2.06	1.91	3.42	2.28	2.93	3.93	3.23	3.07	3.17	3.20	2.64	2.89	2.85	2.84	3.35	2.92	2.83	2.96	2.99	2.99	
11. Cryptocentrus malindiensis (1)	22.24	10.32	15.49	23.31	24.35	23.27	21.61	18.43	16.74	15.01		1.79	1.83	3.12	2.77	2.83	3.64	2.80	3.56	3.26	2.92	2.55	2.55	2.84	3.13	3.62	3.13	3.02	2.91	2.95	2.80	
12. Cryptocentrus maudae (2)	21.46	8.62	20.22	27.25	26.52	30.12	25.28	17.12	16.39	14.54	10.11		2.39	3.23	2.54	3.16	3.76	3.05	3.39	3.49	3.61	2.96	3.05	2.98	3.06	3.98	3.62	3.36	3.05	3.12	3.11	
13. Cryptocentrus nigrocellatus (1)	19.13	12.23	15.73	24.29	24.48	25.53	22.88	17.56	13.97	14.06	13.27	16.06		3.09	2.58	2.93	3.65	2.97	3.37	3.41	3.27	2.97	2.70	3.18	2.72	3.18	2.89	2.61	2.88	2.75	2.99	
14. Cryptocentrus pavoninoides (5)	20.98	25.11	23.37	4.00	19.12	14.85	13.62	25.64	21.06	24.80	23.81	25.22	25.03		3.08	3.01	3.92	3.03	3.30	3.28	3.38	3.40	3.24	2.76	3.48	3.54	3.45	2.86	3.79	3.51	3.31	
15. Cryptocentrus sp. (3)	23.48	16.77	16.45	24.89	25.45	24.62	24.62	14.44	17.84	16.80	20.81	20.17	19.03	25.62		3.48	3.47	3.36	3.21	3.52	3.15	3.59	3.17	2.90	2.96	2.50	2.78	2.82	2.87	2.73	3.61	
16. Cryptocentroides arabicus (5)	23.85	22.23	23.88	24.22	23.88	27.13	24.06	21.67	22.04	25.95	24.10	26.89	23.15	24.67	25.86		3.59	2.86	3.24	3.34	2.96	3.08	3.13	2.99	3.35	3.82	3.36	3.15	3.39	3.22	3.43	
17. Amblyeleotris diagonalis (2)	29.30	25.93	24.88	28.99	28.14	26.46	31.22	24.42	25.36	27.09	27.32	28.14	24.53	29.50	26.28	27.86		2.15	2.90	2.88	2.80	3.39	3.30	3.73	2.75	3.95	3.79	3.92	2.70	2.91	2.92	
18. Amblyeleotris downingi (1)	25.40	20.30	23.03	25.51	26.50	26.80	22.47	24.30	23.67	24.13	22.51	24.26	21.91	26.31	25.61	23.53	14.47		2.52	2.71	2.91	3.45	3.03	3.03	2.89	3.89	3.77	3.47	3.27	3.28	2.94	
19. Amblyeleotris periophthalma (3)	30.27	27.65	30.59	28.40	25.95	27.27	26.96	26.61	27.15	27.40	29.77	29.51	27.70	29.06	27.40	26.02	19.83	17.48		2.93	2.95	3.39	3.78	3.34	3.04	3.66	3.36	3.11	3.45	2.83	3.39	
20. Ctenogobiops feroculus (3)	26.31	25.15	25.66	25.76	26.47	25.15	28.14	24.52	27.16	25.65	26.55	29.65	24.28	27.70	26.93	24.16	20.68	19.61	22.04		2.19	3.56	3.43	3.14	3.59	3.80	3.61	3.57	2.78	3.36	2.99	
21. Ctenogobiops tangaroai (1)	25.66	27.34	24.68	26.87	29.20	27.21	27.96	25.14	24.12	27.85	25.90	28.71	25.26	27.91	26.15	23.91	23.69	20.59	21.34	14.39		2.78	3.30	3.51	3.29	3.40	3.78	3.29	3.02	2.79	2.84	
22. Lotilia sp. (7)	24.99	21.74	21.18	25.59	26.00	25.75	25.36	22.66	20.55	21.44	20.02	23.87	21.56	27.02	25.92	26.13	28.61	28.23	29.97	27.98	25.20		3.20	3.41	2.93	4.26	3.75	3.42	2.94	3.07	3.09	
23. Mahidolia mystacina (1)	23.09	22.12	19.61	26.28	25.69	25.37	20.82	22.74	21.11	23.93	21.35	24.45	20.14	25.59	24.33	27.09	27.62	26.71	29.32	24.84	24.38	23.95		2.78	2.48	3.35	2.99	3.06	2.90	3.16	2.79	
24. Myresina filifer (10)	22.24	24.23	26.96	22.71	23.91	23.88	22.36	24.55	23.52	21.46	25.47	25.00	22.94	23.37	23.85	26.58	29.12	27.05	30.72	25.16	27.46	25.96	23.83		3.27	2.62	2.46	2.49	3.20	3.24	3.42	
25. Psilogobius mainlandi (3)	22.05	22.82	23.93	28.18	27.22	25.44	24.97	23.55	24.39	22.63	21.51	22.93	20.30	28.66	23.71	27.16	23.88	23.43	27.17	28.26	26.37	23.44	20.17	25.33		3.45	3.45	3.42	3.10	2.78	3.31	
26. Stonogobiops medon (4)	23.00	25.17	23.32	23.59	24.13	23.31	28.67	25.94	25.59	23.95	25.83	30.22	23.37	24.72	20.39	27.95	29.47	31.64	30.98	30.48	27.40	31.45	23.19	18.26	26.58		0.86	1.08	3.89	3.33	3.84	
27. Stonogobiops nematodes (1)	21.39	24.18	24.34	24.93	23.89	24.85	28.80	26.01	23.35	22.98	22.72	28.30	22.09	25.81	21.61	26.20	30.30	31.31	29.38	29.77	29.46	30.29	22.24	17.71	25.56	4.06		0.92	3.44	2.98	4.06	
28. Stonogobiops xanthorhinica (1)	18.54	22.75	23.59	22.11	24.64	24.10	26.83	24.17	22.31	21.93	23.34	27.06	20.45	22.26	21.24	24.77	30.11	29.59	27.96	28.94	28.13	28.58	22.44	16.51	26.45	4.66	3.28		3.34	3.26	3.79	
29. Tomiyamichthys lanceolatus (1)	24.42	24.26	27.04	32.01	27.42	31.24	29.71	22.86	25.08	26.88	26.85	27.46	25.74	33.74	24.26	26.63	22.05	26.80	26.34	23.24	22.87	24.87	24.15	27.89	26.43	30.92	29.44	29.01		3.05	3.28	
30. Vanderhorstia mertensi (2)	22.44	25.40	22.88	29.27	30.06	31.71	29.02	23.49	25.27	24.88	25.05	26.44	23.38	29.57	24.56	26.96	22.34	24.45	25.38	28.13	22.54	29.08	27.24	27.60	23.15	25.79	24.71	26.39	25.64		3.10	
31. Gobius niger (Outgroup)	26.91	23.97	23.70	26.46	30.44	24.05	26.38	25.69	24.79	22.94	23.72	25.52	24.34	27.17	30.66	29.15	25.24	24.76	28.95	27.47	24.71	27.15	25.85	30.60	25.76	30.85	32.77	30.59	27.30	28.19		

Finally, the cluster of reef shrimp-gobies reveals two possible misidentifications: (1) Tomiyamichthys lanceolatus, which may be regarded as a Vanderhorstia species (see Fig. 1 in Thacker & Roje, 2011) (2) and Vanderhorstia mertensi, which is shown here based on a single sequence from its invasive population in the Mediterranean Sea. Both putative species in this case are suspected to be the result of a wrong assignment, considering the weak diagnostic characteristics of the genus (Shibukawa & Suzuki, 2004).

Etymology

The new species named after Michael H. Steinhardt in recognition of his immensely important contribution to the establishment and construction of the Steinhardt Museum of Natural History at Tel Aviv University, Israel.

Discussion

As evident from the genetic results of this study (Fig. 3) as well as from the findings of Thacker & Roje (2011), Thacker (2015) and McCraney, Thacker & Alfaro, 2020, the generic status and validity of some shrimp-associated gobies remains to be settled and requires further revisional examinations incorporating additional species and more genetic markers. In the present study, we followed the status of the genera and species as presented by Fricke, Eschmeyer & van der Laan (2021).

The Red Sea is the main origin for over 400 alien species reported from the Mediterranean coast of Israel, among them five goby species (Galil et al., 2020). In the Red Sea the number of shrimp-gobiy species is 23, as featuring in the latest checklist of the Red Sea fishes (Golani & Fricke, 2018). These species belong to eight genera: Amblyeleotris (6 spp.), Cryptocentroides (1 sp.), Cryptocentrus (4 spp.), Ctenogobiops (3 spp.), Lotilia (1 sp.), Psilogobius (1 sp.), Tomiyamichthys (3 spp.) and Vanderhorstia (4 spp.).

Cryptocentrus steinhardti differs from the species of the genera Vanderhorstia, Ctenogobiops, Cryptocentroides and Tomiyamichthys in possessing transverse sensory papillae on the head vs.. longitudinal sensory papillae on the head (Larson & Murdy, 2001; Bogorodsky, Kovačić & Randall, 2011).

The Red Sea species Cryptocentroides arabicus (Gmelin, 1789), which is superficially similar to C. steinhardti, differs from the new species in possessing longitudinal sensory papillae on the head. In addition, C. arabicus differs in possessing a thin dermal crest on top of the head in front of the dorsal fin (Larson & Murdy, 2001) and a restricted gill opening, extending to below pectoral-fin base in Cryptocentroides (Akihito, Hasayoshi & Yoshino, 1984) vs. no dermal crest on top of head and a wide gill opening, reaching to below the preopercular margin, in C. steinhardti (Fig. 2).

Psilogobius spp. differ from the new species in possessing ctenoid scales on the posterior part of the body, lacking pre-opercular pores (Watson & Lachner, 1985) and the presence of thin vertical white lines on side of the body (Larson & Murdy, 2001).

Cryptocentrus steinhardti differs from the Amblyeleotris spp. in possessing pelvic fins completely united with a well-developed fraenum vs. completely separated pelvic fins in Amblyeleotris (Hoese, 1986).

Lotilia spp. differ from the new species in possesing naked predorsal midline and lower scale count along the body (fewer than 53 in Lotilia spp. (Shibukawa, Suzuki & Senou, 2012)).

Thacker, Thompson & Roje (2011) recognized two different clades in this group: one clade contains the genera Amblyeleotris, Ctenogobiops and Vanderhorstia and the other contains Cryptocentrus, Mahidolia, and Stonogobiops. McCraney, Thacker & Alfaro (2020) assigned the species of the genera Amblyeleotris, Ctenogobiops, Vanderhorstia and Tomiyamichthys latruncularius (Klausewitz 1974) to the clade Asterropteryx (together with non-shrimp associated genera Asterropteryx and Gladiogobius) and the other shrimp-goby genera including Tomiyamichthys oni (Tomiyama 1936) to the clade “Cryptocentrus”. Larson & Hoese (2004) suggested, after examining 28 species of Cryptocentrus, that this genus is not monophyletic. This approach was supported by a generic dendrogram, Fig. 6 in McCraney, Thacker & Alfaro (2020), although their “Cryptocentrus” clade contains only ten species of Cryptocentrus. Our findings also show that “Cryptocentrus” is a polyphyletic group (Fig. 3) and includes species of the genera Stonogobiops, Mahidolia, Myersina, Psilogobius and Lotilia. Based on present phylogenetic analysis (Fig. 3) in case of splitting the genus Cryptocentrus into two groups, species which closely related to the type species C. cryptocentrus (Valenciennes, 1837) can apply to the true Cryptocentrus whereas another generic name is required for the remaining group of “Cryptocentrus” species, with unclear position of Lotilia in between C. steinhardti and rest of “Cryptocentrus”. Thus, the relationship of the new species among its congeners and closely-related genera should further studied. The differences between the new species and the species of Psilogobius and Lotilia are described above. Mahidolia spp. differ from C. steinhardti in having fewer than 45 scales along the body (vs. more than 55) and in the absence of an anterior interorbital pore vs. the presence of an interorbital pore (Hoese, 1986). Myersina spp. differ from C. steinhardti in lacking scales on mid nape (Winterbottom, 2002). Stonogobiops spp. differ from the new species in having large vomerine teeth (Winterbottom, 2002) vs. none in the new species.

In light of the morphological characteristics and genetic analyses, we provisionally allocate the new species to the genus Cryptocentrus, despite the COI phylogenetic tree that has appeared to have positioned it within a different clade of genera (Fig. 3).

This genus currently comprises 36 species (Froese & Pauly, 2021). Allen & Randall (2011) distinguished a group of four species characterized by possessing fewer than 70 scales in longitudinal series along the body. They included the following four species in this group: C. caeruleomaculatus (Herre, 1933), C. cyanospilotus Allen & Randall, 2011, C. insignitus (Whitley, 1956) and C. strigilliceps (Jordan & Seale, 1906). The group was then expanded with the descriptions of C. epakros Allen, 2015 (Allen, 2015) and C. altipinna Hoese, 2019 (Hoese, 2019). Two of these species, C. caeruleomaculatus and C. strigilliceps are known from the western Indian Ocean (Froese & Pauly, 2021), but none of these have been reported to date from the Red Sea (Golani & Fricke, 2018).

Cryptocentrus steinhardti differs from all other members of this group, except C. insignitus and C. epakros in possessing cycloid scales only. It differs from C. insignitus in possessing a higher number of scales along the body (50–55 vs. 59–61), the presence of mid predorsal scales (Table 3) and no ocellus on the first dorsal fin. Cryptocentrus epakros differs from C. steinhardti by possessing a lower number of scales along the body (47 vs. 59–61) and fewer transverse scales (12 vs. 19–21).

Table 3 Selected meristic counts of “low scale count group” Cryptocentrus (sensu Allen & Randall, 2011).

Species	LL	2nd D	A	PreD-Mid line	Ctenoid scales on posterior part of body	
Cryptocentrus steinhardti n.sp.	59–61	I + 10–11	I + 9−10	20–21	–	
Cryptocentrus cyanospilotus 1	49–59	I + 10	I + 9	10–13	+	
Cryptocentrus caeruleomaculatus 2	60	I + 10	I + 9	none	+	
Cryptocentrus strigilliceps 3	45–57	I + 10	I + 9	“Predorsal midline and sides scaled to a point just before to just behind posterior preopercular margin” (Hoese, 2019)	+	
Cryptocentrus insignitus 5 , 4	50–55	I + 12	I + 11	None on mid predorsal. Few scattered on napes’ sides and shoulders.	–	
Cryptocentrus altipinna 3	56–65	I + 10	I + 9	none	+	
Cryptocentrus epakros 4	47	I + 10	I + 9	19	−	
Notes:

LL-No. of scale series along the body; 2nd D–No. of spine and segmented rays in second dorsal fin; A-No. of spine and segmented rays in anal fin; PreD-No. pre-dorsal scales.

1 Allen & Randall (2011).

2 Herre (1933).

3 Hoese (2019).

4 Allen (2015).

5 Whitley (1956).

According to Golani & Fricke (2018) four species of Cryptocentrus have been reported from the Red Sea: Cryptocentrus caeruleopunctatus (Rüppell 1830), Cryptocentrus cryptocentrus (Valenciennes 1837), Cryptocentrus fasciatus (Playfair 1867) and Cryptocentrus lutheri Klausewitz 1960. Cryptocentrus steinhardti differs from these four species in lower scale count along the body (59–61 vs. 77–108), lower transverse scale series (19–21 vs. 29–43) and lower number of gill rakers on the lower limb of first arch (8–9 vs. 11–13, including angle’s raker; Table 4).

Table 4 Compression of selected counts of Red Sea species of Cryptocentrus.

	No. of scales in longitudinal series	No. of scale in transverse series	Gill rakers on lower limb of first arch (including angle’s raker)	
Cryptocentrus steinhardti n. sp.	59–61	19–21	8–9	
Cryptocentrus caeruleopunctatus (Ruppell 1830)	78–941	301	111	
Cryptocentrus cryptocentrus (Valenciennes 1837)	80–1082; 70–901	34–351	122; 131	
Cryptocentrus fasciatus (Playfair 1867)	773; 81–922	30-432	123; 12–132	
Cryptocentrus lutheri Klausewitz 1960	98–1021	291	111	
Notes:

1 Goren (1979).

2 Polunin & Lubbock (1977).

3 Randall & Goren (1993).

The finding of a new Indo-Pacific invasive species in the Mediterranean prior to its discovery in the Indo-Pacific Ocean or Red Sea is an unusual event, although other such cases have been previously documented. The snapping shrimp Alpheus migrans Lewinsohn & Holthuis, 1978, which belongs to an Indo-Pacific species group, was first described from the Mediterranean (Lewinsohn & Holthuis, 1978); the jellyfish Marivagia stellata Galil & Gershwin, 2010 was described from the Mediterranean and later also reported from India (Galil, Kumar & Riyas, 2013); the flounder Arnoglossus nigrofilamentosus Fricke, Golani & Appelbaum-Golani, 2017 (Fricke, Golani & Appelbaum-Golani, 2017), which is probably a Red Sea species; the goby Hazeus ingressus Engin, Larson & Irmak, 2018, which belong to an Indo-Pacific genus, was discovered in the Mediterranean (Engin, Larson & Irmak, 2018) and later was found in Abu Dabab, Egypt, Red Sea (Bogorodsky, Suzuki & Mal, 2016), and the jellyfish Rhopilema nomadica Galil, Spanier & Ferguson, 1990 (Galil, Spanier & Ferguson, 1990) that was described on the basis of types from the Mediterranean although it is an Indo-Pacific species.

Finding the new shrimp-associated goby, however, which is also the third such goby to be documented as an invasive species in the Mediterranean (after Vanderhorstia mertensi and Cryptocentrus caeruleopunctatus) raises the question of its possible symbiosis with an alpheid shrimp. Since this taxon of gobies possesses either an obligatory or facultative association with shrimp (Lyons, 2013), its pairing with one of the ca. 20 candidate species of alpheid shrimps from the Mediterranean and the Red Sea (Karplus, 2014) can be a key factor for its survival and population establishment success in the invaded territory. Unfortunately, as the catch of C. steinhardti in this study was not associated with any shrimp species, the question of its possible symbiosis in the Mediterranean remains open and in need of further observations.

Last, Cryptocentrus steinhardti was collected during the night and at depths of 60 to 80 m. Finding this species during the period of dark and below the depth limits of recreational diving could be an additional reason for overlooking this species and its possible shrimp associates in its native origin.

Supplemental Information

Supplemental Information 1 Information for the primers used for PCR and sequencing in this study.

Click here for additional data file.

Supplemental Information 2 BOLD information for COI sequences of all available shrimp-associated gobies used for the phylogenetic analysis in this study (n = 111).

Click here for additional data file.

We thank Mr. O. Rittner for the photographs of the fish and Ms. N. Paz for editing the manuscript. We also thank B. Rinkevich (IOLR) and his dedicated staff for long-time assistance in the molecular analyses. Last, we thank S. Bogorodsky, H. Larson and an anonymous reviewer that greatly improve the quality of the manuscript.

Additional Information and Declarations

Competing Interests

Author Contributions

Data Availability

New Species Registration

The authors declare that they have no competing interests.

Menachem Goren conceived and designed the experiments, performed the experiments, analyzed the data, prepared figures and/or tables, authored or reviewed drafts of the paper, contributed the taxonomic description, and approved the final draft.

Nir Stern conceived and designed the experiments, performed the experiments, analyzed the data, prepared figures and/or tables, authored or reviewed drafts of the paper, contributed the molecular and systematic analyses, and approved the final draft.

The following information was supplied regarding data availability:

The data is available at BOLD: BIM769-20 for the holotype and BIM534-17 and BIM770-20 for the two paratypes.

The specimens are stored at the fish collection in the Steinhardt museum of natural history in Tel Aviv University, Israel.

Holotype museum voucher-SMNH P-16037

Paratype voucher-SMNH P-14556.

The following information was supplied regarding the registration of a newly described species:

Publication LSID: urn:lsid:zoobank.org:pub:B5279F4D-F5BC-454D-9ED8-3E2A13C69EAE

Cryptocentrus steinhardti n. sp. LSID: urn:lsid:zoobank.org:act:D16A267F-AF39-4E39-BBE2-3F4014E941CA.

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
