# Peer review of "Cryptocentrus steinhardti (Actinopterygii; Gobiidae): a new species of shrimp-goby, and a new invasive to the Mediterranean Sea"

_PeerJ, doi:10.7717/peerj.12136_

## Round 0.1 · original submission · Major Revisions

The three reviewers are experienced ichthyologists with in-depth knowledge of this group of fish. You will see that two reviewers were not convinced that the specimens represent a new species. Two reviewers provided detailed comments on the original version of your text. I believe that a lot of additional work will be needed to convince them, and I will make a positive decision only when they are fully convinced.

Reviewer 1 ·

Basic reporting

The paper has several grammar and spelling errors, particularly some in the scientific names. There is not enough background or context given to support the conclusions, and not enough literature surveyed. The figures are good, and the data has been shared, but the hypothesis that the described individuals represent a new species of Cryptocentrus is not sufficiently supported.

Experimental design

The research is within the aims and scope of the journal, and the question is well defined, but the investigation is not rigorous enough and the methods need more detail.

Validity of the findings

The data are clearly given but they are not robust and the conclusions are not well-supported.

Additional comments

This contribution describes morphological and genetic characteristics of three fish individuals trawled from the eastern Mediterranean and assigned to the shrimp goby genus Cryptocentrus. The authors describe the species as new and postulate that it arrived in the Mediterranean from the Red Sea via the Suez Canal. This dispersal pattern is not unknown, there are several examples of such Lessepsian migrants including two other shrimp gobies (Vanderhorstia mertensi and Cryptocentrus caeruleopunctatus), but it is unusual to have a new species that presumably arrived from the Red Sea described first from the Mediterranean. Unfortunately, the authors do not make sound arguments for any of their claims, and a more comprehensive and careful consideration of the characters and diversity of shrimp gobies would be needed to definitively describe these individuals as a new species. The three individuals from the trawl are all damaged, and so the authors use DNA sequences to assist with identification. However, their methods are incompletely described and the resultant phylogeny has many very poorly supported nodes, making their phylogenetic conclusions unreliable.

The morphological argument that the specimens represent a new Cryptocentrus species rests on first establishing that it is a Cryptocentrus, and then on determining that it is new. As the authors discuss, the generic diagnoses for shrimp goby genera are not always reliable, and in this case they use molecular data to place the new individuals among other goby species. They sequence the mitochondrial COI and cytb genes, although only the COI is used for phylogenetic reconstruction. The description of the molecular methods is very brief and inconsistent: is the tree an ML tree or Bayesian? The figure caption says ML but the description of replications given in the methods is more consistent with a Bayesian analysis. There are almost no details given about how the sequences were alighned, concatenated, how models were identified, and how the trees were run. Why wasn't cytb included in the phylogeny? Other cytb data for Cryptocentrus are available. Much more detail is needed here, but unfortunately, it may not result in a better phylogeny. The phylogeny given is very poorly supported and it's really not possible to say much more than the new individuals are probably part of the Cryptocentrus lineage.

The authors also need to provide much more, and more accurate, morphological comparisons between the new individuals and known species. For instance, they discuss the character of transverse (rather than longitudinal) papillae rows on the head, and state that the Asterropteryx lineage (including shrimp goby genera such as Amblyeleotris, Ctenogobiops and Vanderhorstia) is characterized by longitudinal papillae rows, while the Cryptocentrus lineage has transverse papillae rows, and they cite Thacker et al. 2011. That paper says the exact opposite of what the authors describe: transverse papillae rows are shared by Amblyeleotris, Cryptocentrus, Myersina and Stonogobiops, they are not clade-specific and cannot be used to assign species to shrimp goby lineages. This is a pretty large mistake and makes me question the accuracy of the rest of the morphological comparisons. The authors go on to argue that the new individuals are distinguished from species of Cryptocentroides in having a wide gill opening, extending to below the preopercular margin, as opposed to a restricted gill opening, extending to below the pectoral fin base. Those don't sound very different, an illustration would be very helpful. Is that the only character that diagnoses Cryptocentroides? The new individuals are separated from Myersina by the degree of nape scalation. How variable is that, and are there other characters typical of Myersina? What distinguishes Cryptocentrus from Vanderhorstia or Tomiyamichthys, and which of those characters are present in the new individuals? This is covered briefly but needs expansion. The authors provide little detail and no citations; this section needs to be greatly expanded so that the generic assignment is solidly supported.

Finally, the authors compare the new individuals to Cryptocentrus species possessing fewer than 70 lateral scales (most Cryptocentrus have more), but none of those species are present in the Red Sea. The comparisons and differential diagnoses need to be expanded and Red Sea species must be considered; many species have wide ranges of lateral scales, the new individuals might be within the range of intraspecific variation for some Red Sea Cryptocentrus. I'd recommend looking carefully at Cryptocentroides arabicus and Cryptocentrus fasciatus. It is also possible that the new individuals are from previously known Lessepian migrant species Cryptocentrus caeruleopunctatus, there is so little detail given, and no discussion of variation, that it is impossible to know how the new individuals compare.

·

Basic reporting

All comments are on the attached pdf.
The manuscript needs work as it relies almost entirely on genetic samples of 3 specimens (COI) and almost no morphological comparison. It may indeed be a new species but authors need to convince the reader properly. They have not demonstrated that their "comprehensive morphological and anatomical examinations" show this.

Experimental design

Plenty of genetic info but considerably lacking in the "comprehensive morphological and anatomical" information it is supposed to contain. See comments in pdf.

Validity of the findings

See comments in pdf.

Additional comments

Please convince the reader that this really is a good species with good morphological comparison with its congeners in the western Indian Ocean, especially the Red Sea.

·

Basic reporting

No addition comments.

Experimental design

In general good article with another new species described from non-native area.
All corrections and suggestions are visible through tracking changes in 3 files.
Some corrections are needed.

Validity of the findings

Authors found a new species for science without doubt.
But its and Lotilia relationships to other Cryptocentrus species were not resolved by present analysis and an additional study is needed, making question of generic name of part of Cryptocentrus species opened.

Additional comments

The description of new species is provided in shortened style but enough to see details of morphological characters. But some mistakes are present, for example authors wrongly assigned shrimp-gobies genera to Asterropteryx "lineage", i.e. the genus is not associated with shrimps. Authors need to be more careful with such conclusions.
All corrections within Word files.

---

## Round 0.2 · Minor Revisions

You will see that all three reviewers are now accepting the new species, but they still require a number of minor modifications in the text, and in at least one figure.

Most comments were provided as annotations on your manuscript.

Reviewer 1 ·

Basic reporting

In this revision, the authors have made many improvements over the previous version, including expanded morphological comparisons and improvements to the text. The molecular analysis is unchanged, but is better explained and the shortcomings are better acknowledged (the name Myersina is still misspelled in Figure 3, and Ctenogobiops is misspelled on line 180). Overall, the writing is clearer and it reads well.

Experimental design

The research is within the aims and scope of the journal, and the question is well defined. The MS has been updated in response to reviews with additional data and comparisons to other Cryptocentrus species, both the congeners in the Red Sea and all of the species in the Cryptocentrus "low scale count" group. The Discussion has also been expanded with more comparisons to other shrimp goby genera and some discussion of their diagnostic characters (or lack thereof). These are excellent additions and greatly strengthen the argument that the species is new.

Validity of the findings

The arguments for the novelty of C. steinhardti are good, much stronger now with the added comparative material. The molecular results are unchanged and are still basically inconclusive, but the supporting text makes it clear that Cryptocentrus is in need of revision and that the phylogeny is not particularly robust.

Additional comments

The authors have added significant material that greatly strengthens the paper, well done.

·

Basic reporting

no additional comments.

Experimental design

no additional comments.

Validity of the findings

no additional comments.

Additional comments

The article after first review was improved but there are some places where additional corrections are needed. All my corrections, comments and suggestions are visible in attached Word file.

---

## Round 0.3 · accepted · Accept

One reviewer was kind enough to edit the list of references, but it is my understanding that this is usually done by the staff of this journal. In view of the comments by the three reviewers, I believe this manuscript should be now accepted and published.

·

Basic reporting

See below

Experimental design

See below

Validity of the findings

See below

Additional comments

I have read the revision materials and am happy to Accept the manuscript

·

Basic reporting

In general no additional comments except for formatting.

Experimental design

no comments

Validity of the findings

no comments

Additional comments

Few corrections (mostly commas) were made through MS and visible in Corrections.
I was sure authors can check Instructions of preparation of References according journal format, but authors omitted it.
Therefore I made formatting of references following journal rules.
All corrections are visible in MS.